# Cold Atmospheric Plasma Activates Selective Photothermal Therapy of Cancer

**DOI:** 10.3390/molecules27185941

**Published:** 2022-09-13

**Authors:** Jiamin Qin, Jingqi Zhang, Guojuan Fan, Xiaoxia Wang, Yuzhong Zhang, Ling Wang, Yapei Zhang, Qingfa Guo, Jin Zhou, Weifen Zhang, Jinlong Ma

**Affiliations:** 1School of Pharmacy, Weifang Medical University, Weifang 261053, China; 2Department of Skin, Weifang Hospital of Traditional Chinese Medicine, Weifang 261000, China; 3Department of Biomedical Engineering, Michigan State University, East Lansing, MI 48824, USA; 4College of Chemical Engineering and Environmental Chemistry, Weifang University, Weifang 261061, China; 5Collaborative Innovation Center for Target Drug Delivery System, Weifang Medical University, Weifang 261053, China; 6Shandong Engineering Research Center for Smart Materials and Regenerative Medicine, Weifang Medical University, Weifang 261053, China

**Keywords:** cold atmospheric plasma, selective photothermal therapy, gold nanostars, cancer treatment

## Abstract

Due to the body’s systemic distribution of photothermal agents (PTAs), and to the imprecise exposure of lasers, photothermal therapy (PTT) is challenging to use in treating tumor sites selectively. Striving for PTT with high selectivity and precise treatment is nevertheless important, in order to raise the survival rate of cancer patients and lower the likelihood of adverse effects on other body sections. Here, we studied cold atmospheric plasma (CAP) as a supplementary procedure to enhance selectivity of PTT for cancer, using the classical photothermic agent’s gold nanostars (AuNSs). In in vitro experiments, CAP decreases the effective power of PTT: the combination of PTT with CAP at lower power has similar cytotoxicity to that using higher power irradiation alone. In in vivo experiments, combination therapy can achieve rapid tumor suppression in the early stages of treatment and reduce side effects to surrounding normal tissues, compared to applying PTT alone. This research provides a strategy for the use of selective PTT for cancer, and promotes the clinical transformation of CAP.

## 1. Introduction

Cancer is one of the most severe human diseases [1]. According to current statistics, over 19 million people worldwide have been diagnosed with cancer, and almost 10 million cancer deaths occurred in 2020. By 2040, new cases and deaths are expected to reach approximately 28 million and 16 million, respectively [2,3,4]. Despite significant medical, scientific, and technological advances, cancer is still a disease with limited treatment approaches [5]. Traditional cancer therapies, including surgery, radiation therapy, and chemotherapy, have inherent limitations, such as multidrug resistance, low therapeutic efficiency, and high recurrence [1,6,7,8,9]. Photothermal therapy (PTT) can be used to induce tumor cell apoptosis or necrosis by hyperthermia, thus inhibiting tumor growth [10,11,12,13,14]. PTT has several advantageous attributes for cancer therapy: simple operation, short treatment time, noninvasiveness, and rapid recovery. The ultimate goal of cancer treatment is to destroy tumor tissues without damaging normal tissues. However, PTT has the limitation of being poorly selective as a result of the systemic distribution of photothermal agents throughout the body and the imprecise exposure of lasers. This limitation can cause serious side effects on normal tissues surrounding the tumors, limiting the application of PTT [11,15,16,17,18].

Cold atmospheric plasma (CAP) is a highly energized gas that is composed of numerous biologically active factors where the ion temperature is close to room temperature. They may act synergistically on the treated materials and tissues, and may have various biologic effects [6,19,20,21]. CAP has numerous biologically active components, of which the basic components are various reactive oxygen species (ROS) and reactive nitrogen species (RNS), which are formed in reactions with molecules (oxygen, nitrogen, and water) present in the ambient air [22]. Research has suggested that CAP provides the effect of selectively targeting cancer cells, as cancer cells are more sensitive to CAP treatment than healthy cells [22,23,24,25,26,27]. This anticancer selectivity is related to the concentration of reactive oxygen and nitrogen species (RONS) generated by CAP [28]. CAP can target and kill cancer cells by increasing oxidative stress levels. RONS also can induce cancer immunogenic cell death, and then release tumor-associated antigens. In tumor-draining lymph nodes, mature dendritic cells give antigen peptides to T cells, producing cytotoxic T cells. The subsequent T-cell-mediated immune response can inhibit tumor recurrence and growth [29]. In addition, the higher aquaporin density on the cell membrane causes a higher inflow of hydrogen peroxide to kill cancer cells [8,30,31]. However, the direct killing ability of the tumor of CAP is weak, and using it as a supplementary procedure or as a precautionary measure may be its primary application strategy [20,32]. Therefore, we considered whether greater selectivity of PTT for cancer could be realized by using CAP as a supplementary procedure. The combination of PTT with CAP at lower power achieved a similar killing effect to that of using high power irradiation alone.

In this study, we used CAP as a supplementary procedure to enhance the selective PTT of cancer (Figure 1). In order to verify the feasibility of this strategy, the classical photothermic agent’s gold nanostars (AuNSs) were used here. AuNSs were modified with folic acid (FA) to increase the enrichment in the tumor site by targeting the overexpressed FA-receptors of tumor cells. Collectively, we believe that the combination of CAP and PTT can increase the selectivity of PTT and promote the clinical transformation of CAP.

## 2. Results and Discussion

### 2.1. Synthesis and Characterization

Gold nanostars (AuNSs) were prepared by reducing HAuCl_4_ with HEPES, a dark blue transparent solution. The hydrated particle size of the AuNSs was about (24.49 ± 0.25) nm (Figure 1a). From Figure 1b, we can find that the structure under TEM resembles a star, which proves the successful preparation of the AuNSs. AuNSs modified with folic acid (FA) can actively target tumor cells because the folate receptor (FOLR), a tumor-associated protein, is overexpressed on the surfaces of various cancer cells [33,34,35,36]. La-PEG-FA was modified on the surface of AuNSs by coordination with Au-S of the modified material [37]. The hydrated particle size of AuNSs@FA is 43.08 ± 0.19 nm (Figure 1a), which proves that La-PEG-FA had been successfully modified on the AuNSs. The changes in the surface plasmon resonance absorption peak were also traced in order to study FA modification further. As shown in Figure 1c, the surface modification of the AuNSs results in a red shift compared with the unmodified AuNSs (704.8 nm), indicating successful surface modification. Moreover, the zeta potential change in the AuNSs further proves the success of FA modification [38] (Figure 1d).

### 2.2. Photothermal Effect of AuNSs@FA

With a high absorption-to-scattering ratio in the NIR region, AuNSs are emerging rapidly as an optimal photothermal conversion material, and are being used in photothermal therapy (PTT) [39,40]. Here, the photothermal conversion abilities of AuNSs@PEG and AuNSs@FA under near-infrared illumination were investigated. From Figure 2a,b,d,e, we can see that the photothermal conversion abilities, with or without CAP treatment of both AuNSs@PEG and AuNSs@FA, are concentration dependent. Interestingly, the temperatures of both AuNSs@PEG and AuNSs@FA ([Au] concentration: 100 mg/mL) show an obvious increase (ΔT ≥ 5 °C) after CAP treatment (Figure 2c,f). CAP is an effective material surface modification tool. It can emit high energy and charged particles onto the treated surface, resulting in the breakage of covalent bonds and initiating many chemical reactions [41]. Therefore, we speculate that the surface properties of the AuNSs changed after CAP treatment, and consequently their photothermal conversion ability increased [42,43,44]. Photothermal conversion capability under different powers was also evaluated. From Figure 3a,b,d,e, we can see that the photothermal conversion temperatures of AuNSs@PEG and AuNSs@FA were raised as the laser power increased. As shown in Figure 3c,f, the temperatures of the AuNSs, modified with or without FA, were also significantly higher after CAP treatment under different powers. In addition, the thermal stabilities of AuNSs@PEG and AuNSs@FA were evaluated. After five repeated laser irradiation-cooling cycles, their photothermal conversion abilities were nearly unaffected. Importantly, after CAP treatment, the thermal stabilities of AuNSs@PEG and AuNSs@FA were not affected (Appendix A). Therefore, AuNSs@FA was used for the next study as a result of its excellent photothermal conversion ability and good photothermal stability.

### 2.3. Biocompatibility

The size stability of nanoparticles is an essential parameter of their biological application and storage [45]. The size of AuNSs@FA in PBS and RPMI remained stable after incubation at 4 °C and 37 °C for 12h (Figure 4a,b). The size also remained stable in PBS after incubation at 4 °C and 37 °C for 1 week, which indicated that the particles had good stability (Figure 4c).

The biological safety of the AuNSs@FA carrier was examined, an essential prerequisite in biomedical applications. We accessed via hemolysis assays, L929 cells cytotoxicity, and organic safety. From Appendix A, the hemolysis rates were all less than 5%, indicating no hemoglobin release occurred. L929 cells were selected to investigate the cytotoxicity of AuNSs@FA. The MTT assay results showed that the cells demonstrated good activity, even as the concentration of AuNSs@FA reached 1.0 mM (Figure 4d). A live/dead assay for the L929 cells was also measured to better illustrate the cytotoxicity of AuNSs@FA for normal human cells under different treatments. Appendix A shows great activity with different treatments, indicating that AuNSs@FA has almost no cytotoxicity to normal human cells.

In addition to the excellent biocompatibility reports of the hemolysis test and the cytotoxicity evaluation, the organic safety of AuNSs@FA in healthy mice was further investigated. The major organs were collected after injecting PBS or 100 μL of AuNSs@FA at an Au concentration of 60 mM, and breeding for 10 days. Figure 4e shows that no damage occurred in organs injected with AuNSs@FA compared with those that received PBS. Combined with the in vitro and in vivo biosafety evaluations, the excellent biosafety of AuNSs@FA illustrates the potential to be used in delivery.

### 2.4. In Vitro PTT Effects

The MTT assay of the 231cells was used to examine the tumor cell inhibition ability of AuNSs@FA via cell killing experiments. Figure 5a shows that with or without CAP treatment, AuNSs@FA has a better killing effect than AuNSs modified without FA in the concentration range of 0.2–1 mM, indicating a targeting ability of FA to cancer cells. AuNSs@FA with laser irradiation combined with CAP treatment has an excellent killing effect. In order to identify whether CAP could decrease the effective power of PTT, an MTT assay of the 231cells under high power was also evaluated (Figure 5b). Under the same laser irradiation time (3 min), the viability of 231 cells under higher power (1.5 W/cm^2^) and CAP treatment decreased more dramatically (Figure 5c,d). It is worth noting that the cell lethalities of AuNSs@FA+CAP+0.5 W/cm^2^ and AuNSs@FA+1.5 W/cm^2^ have similar therapeutic effects. The ROS and RNS generated by CAP were responsible for the cytotoxic and reactive species produced by CAP. They can directly induce DNA damage and cell cycle arrest, resulting in the apoptotic signaling of tumor cells [1,46]. Compared with high-power laser irradiation, low-power laser irradiation treatment is safer for normal cells. Therefore, PTT under lower power with the assistant treatment of CAP has a similar therapeutic effect as higher power PTT, indicating that CAP could improve the selection of PTT.

A live/dead assay was used to better evaluate, visually, the combined efficiency of CAP and PTT. For ive/dead double staining (Figure 5e), the 231 cells were treated with different powers and combined with CAP. Then, they were stained with AM/PI and observed under an inverted fluorescence microscope. The diagram shows that AuNSs@FA have a better killing effect than AuNSs modified without FA at the same concentration. Similar dead cells with red fluorescence under AuNSs@FA+CAP+0.5 W/cm^2^ and AuNSs@FA+1.5 W/cm^2^ were observed.

### 2.5. In Vivo Photothermal Treatment

In order to examine the in vivo tumor-targeting ability and distribution of AuNSs@FA, a tumor-bearing mouse model was established for in vivo imaging by using an NIRF imaging system by modifying Cy 5.5. From Figure 6a, the tumor fluorescence intensity of Cy5.5-AuNSs@FA increased gradually and reached a peak at 6 h. Compared with the group of AuNSs modified without FA, the tumor site in the AuNSs@FA treatment group exhibited stronger fluorescence at 6 h; this illustrates that the modification of FA can improve the tumor-targeting ability of nanoparticles [47]. The fluorescence signals in tumor sites gradually weakened after 6 h, illustrating that AuNSs@FA was well metabolized in the body. In order to further prove the conjecture, the heart, liver, spleen, lung, kidney, and tumor at 6 h and 48 h were harvested and measured (Figure 6b). Strong fluorescence intensity was found in the tumor sites at 6 h. The fluorescence signals in each organ decreased significantly and were weak at 48 h, which indicated that the gold modified with Cy5.5 was effectively metabolized.

Based on the above in vivo imaging experiments, 6 h after injection was chosen as the optimum time point for laser irradiation treatment. In order to access the characteristics of AuNSs@FA in vivo, the temperature changes after 808-nanometer irradiation were monitored by an infrared thermal imaging camera in the process. From Figure 6c,d, the tumor temperature of the AuNSs@FA group reached 46 °C, compared with 36.6 °C for with PBS-injected group, at a power of 1.0 W/cm^2^. From Figure 6e,f, the local tumor temperature of the AuNSs@FA group could increase by more than 14 °C after laser irradiation at 1.5 W/cm^2^ for 5 min. Moreover, the temperature of AuNSs@FA was higher than AuNSs modified without FA under different powers, illustrating that FA has a targeting ability to the tumor site. Importantly, there was no significant change in skin surface temperature before and after CAP treatment, which cannot cause supernumerary thermal damage [48].

Based on the effective accumulation of AuNSs@FA, tumor-bearing mice were used to further study its antitumor efficacy in the body. During the treatment period, tumor sizes were measured with digital calipers (Appendix A). Figure 7a shows that the volume growth of AuNSs@FA-injected tumors was significantly suppressed after CAP treatment and laser irradiation. The AuNSs@FA+1.5 W/cm^2^ group and the AuNSs@FA+CAP+1.0 W/cm^2^ group showed the best therapeutic effects and had similar suppression effects. The lower power irradiation combined with CAP has a similar tumor suppression effect compared with higher power irradiation alone, testifying to an excellent synergistic treatment effect. Furthermore, the therapeutic effects of multiple treatments were proved through H&E staining (Figure 7b). Larger intercellular spaces and more extensive necroses were discovered in tumor tissue sections in the AuNSs@FA+1.5 W/cm^2^ group and AuNSs@FA+CAP+1.0 W/cm^2^ group, further confirming their better treatment efficacy. Moreover, the H&E staining section results of major organs also demonstrated that there was almost no obvious tissue damage observed among all groups (Appendix A), and the weight of tumor-bearing mice showed almost no change during the treatment process (Appendix A). As shown in Figure 7c, the H&E slice section of the skin with AuNSs@FA+1.5 W/cm^2^ treatment reveals inflammation, but the AuNSs@FA+CAP+1.0 W/cm^2^ treatment group shows no inflammation in the H&E slice section of the skin. This illustrates that the combination of PTT and CAP can reduce the required irradiation power, as well as reduce inflammation in skin tissue, to achieve selective treatment.

### 2.6. In Vivo Experiments

(1)Establishment of animal tumor model

The BALB/c mice (female, 4–6 weeks) were purchased from Jinan Peng Yue Company (Jinan, China). 4T1 (mouse breast cancer) cells were used to establish the tumor model. All of the experimental procedures were approved by the Animal Care and Use Committee of Weifang Medical University. The right hind limbs of mice were subcutaneously injected with a 100-microliter mixture of PBS suspending 4T1 cells (1 × 10^6^). When the tumor volume reached approximately 100 mm^3^, the tumor model was obtained.

(2)Histological examinations

The BALB/c mice (female, 4–6 weeks, body weight: 18–20 g) were randomly divided into two groups (*n* = 3). AuNSs@FA or saline was injected into BALB/c mice via intraperitoneal injection. The body weights of mice were recorded daily for 10 days, and major organs (heart, liver, spleen, lung, and kidney) were collected on day 10. After staining with hematoxylin & eosin (H&E), sectioned tissue was analyzed under a microscope.

(3)In vivo biodistribution analysis

The biodistributions of AuNSs@PEG and AuNSs@FA were evaluated in mice bearing tumors using the NIRF small animal imaging system (PE IVIS Spectrum). A volume of 150 μL of AuNSs@PEG- and AuNSs@FA-modified Cy5.5 was intraperitoneally injected into mice. Whole-body optical imaging was recorded at the following given time intervals: 2, 6, 12, 24, and 48 h. After 6 h and 48 h of processing, the mice were sacrificed, and the imaging information of Cy5.5 in the primary organs and tumors was observed.

(4)In vivo photothermal imaging of tumor

The BALB/c mice (female, 4–6 weeks, body weight: 18–20g) bearing tumors were randomly divided into 3 groups (*n* = 5), saline, AuNSs@PEG, and AuNSs@FA were injected into tumor-bearing mice via intraperitoneal injection. Six hours after injection, the mice were anesthetized with a 10% chloral hydrate solution (400 mg/kg). The tumor sites were irradiated by an 808-nanometer laser (1 and 1.5 W/cm^2^, 5 min), alone or combined with CAP. At the same time, the temperature of the tumor was recorded using a thermal infrared camera every 0.5 min. Industrial-grade Ar was used as a carrier gas in CAP. The flow rate was 2 L/min, and the treatment time was 1.5 min.

(5)In vivo antitumor therapy

When the tumor volume reached approximately 100 mm^3^, the mice were randomly divided into eight groups (*n* = 5). PBS and AuNSs@FA were intraperitoneally injected into the mice. Then, eight groups were treated with PBS, PBS+CAP, PBS+1.5 W/cm^2^, PBS+CAP+1.5 W/cm^2^, AuNSs@FA+1.5 W/cm^2^, AuNSs@FA+CAP, AuNSs@FA+1.0 W/cm^2^, and AuNSs@FA+CAP+1.0 W/cm^2^. The tumor sites were irradiated by an 808-nanometer laser (1 and 1.5 W/cm^2^), 6 h after the intraperitoneal injection. During this period, the body weights of mice were recorded daily. After 14 days of treatment, the tumors were collected for photography. The skin and tumors of mice were stained with H&E, and histological analyses were performed using a pathology section scanner (Pannoramic MIDI FL, Hungary).

Statistical analyses: All results are expressed as means ± standard deviations (SDs) and represent repeated experiments. Differences between the results were conducted with *t*-tests.

(6)Establishment of animal tumor model

The BALB/c mice (female, 4–6 weeks) were purchased from Jinan Peng Yue Company (Jinan, China). 4T1 (mouse breast cancer) cells were used to establish the tumor model. All of the experimental procedures were approved by the Animal Care and Use Committee of Weifang Medical University. The right hind limbs of mice were subcutaneously injected with a 100-microliter mixture of PBS suspending 4T1 cells (1 × 10^6^). When the tumor volume reached approximately 100 mm^3^, the tumor model was obtained.

(7)Histological examinations

The BALB/c mice (female, 4–6 weeks, body weight: 18–20 g) were randomly divided into two groups (*n* = 3). AuNSs@FA or saline was injected into BALB/c mice via intraperitoneal injection. The body weights of mice were recorded daily for 10 days, and major organs (heart, liver, spleen, lung, and kidney) were collected on day 10. After staining with hematoxylin & eosin (H&E), sectioned tissue was analyzed under a microscope.

(8)In vivo biodistribution analysis

The biodistributions of AuNSs@PEG and AuNSs@FA were evaluated in mice bearing tumors using the NIRF small animal imaging system (PE IVIS Spectrum). A volume of 150 μL of AuNSs@PEG- and AuNSs@FA-modified Cy5.5 was intraperitoneally injected into mice. Whole-body optical imaging was recorded at the following given time intervals: 2, 6, 12, 24, and 48 h. After 6 h and 48 h of processing, the mice were sacrificed, and the imaging information of Cy5.5 in the primary organs and tumors was observed.

(9)In vivo photothermal imaging of tumor

The BALB/c mice (female, 4–6 weeks, body weight: 18–20g) bearing tumors were randomly divided into 3 groups (*n* = 5), saline, AuNSs@PEG, and AuNSs@FA were injected into tumor-bearing mice via intraperitoneal injection. Six hours after injection, the mice were anesthetized with a 10% chloral hydrate solution (400 mg/kg). The tumor sites were irradiated by an 808-nanometer laser (1 and 1.5 W/cm^2^, 5 min), alone or combined with CAP. At the same time, the temperature of the tumor was recorded using a thermal infrared camera every 0.5 min. Industrial-grade Ar was used as a carrier gas in CAP. The flow rate was 2 L/min, and the treatment time was 1.5 min.

(10)In vivo antitumor therapy

When the tumor volume reached approximately 100 mm^3^, the mice were randomly divided into eight groups (*n* = 5). PBS and AuNSs@FA were intraperitoneally injected into the mice. Then, eight groups were treated with PBS, PBS+CAP, PBS+1.5 W/cm^2^, PBS+CAP+1.5 W/cm^2^, AuNSs@FA+1.5 W/cm^2^, AuNSs@FA+CAP, AuNSs@FA+1.0 W/cm^2^, and AuNSs@FA+CAP+1.0 W/cm^2^. The tumor sites were irradiated by an 808-nanometer laser (1 and 1.5 W/cm^2^), 6 h after the intraperitoneal injection. During this period, the body weights of mice were recorded daily. After 14 days of treatment, the tumors were collected for photography. The skin and tumors of mice were stained with H&E, and histological analyses were performed using a pathology section scanner (Pannoramic MIDI FL, Hungary).

Statistical analyses: All results are expressed as means ± standard deviations (SDs) and represent repeated experiments. Differences between the results were conducted with *t*-tests.

## 3. Experimental

### 3.1. Materials

Materials: α-lipoic acid (LA, 99%), chloroauric acid trihydrate (99%), 3-(4,5-dimethyl-2thiazolyl)-2,5-diphenyl-2-H-tetrazolium bromide (MTT), HO-PEG_2K_-OCH_3_, 4′,6-diamidino-2-phenylindole dihydrochloride (DAPI, 98%), and dimethyl sulphoxide (DMSO, 99%) were purchased from Shanghai Dibai Chemical Reagent Co. (Shanghai, China). 1-Ethyl-(3-dimethylaminopropyl) carbodiimide hydrochloride (EDC, 98%) and 4-dimethylaminopyridine (DMAP, 99%) were purchased from Shanghai Aladdin Biochemical Technology Co., Ltd. NH_2_-PEG_2K_-NH_2_ was purchased from Seebio Biotech (Shanghai, China) Co., Ltd. Calcein-AM/PI double stain kit was purchased from Shanghai Yisheng Biotechnology Co., LTD (Shanghai, China). AnnexinV FITC apoptosis assay kit was purchased from BD (San Diego, CA, USA). Cy-5.5-NHS was purchased from Xian RuiXi Biological Technology Co., Ltd (Xian, China). 4T1 cell lines, 231 cell lines, and L929 cell lines were obtained from American Type Culture Collection (ATCC). Other chemicals were commercially available and used without further purification.

### 3.2. Preparation of Nanoparticles

(1)Preparation of AuNSs

A volume of 100 µL of chloroauric acid trihydrate solution (10 mg/mL) was added to 100 mM Hepes solution (6 mL, pH 7.4) under a quick stir until it was fully mixed and stored for 1 h in a refrigerator.

(2)Synthesis of La-PEG

A quantity of 2.6 g of lipoic acid was added to a 100-milliliter dichloromethane solution of 5 g of CH_3_O-PEG_2K_-OH (Mw = 2000). After full mixing, 2.248 g of EDCI and 1.5 g of DMAP were added in turn. The mixed solution was appropriately stirred at room temperature for 3 days. After the reaction, a rotary evaporator was used to remove dichloromethane, and the crude product was obtained. Then, the crude product was dissolved in 50 mL of DMSO, and 50 mL of purified water was added. The mixed solution was dialyzed against water for 3 days in dialysis tubing (MWCO 2000). Then, a white flocculation product was obtained after freeze-drying.

(3)Synthesis of La-PEG-FA

An amount of 56.3 mg of folic acid and 42.925 mg of EDCI were dissolved in 25 mL of DMSO. The mixed solution was properly stirred at 35 °C for 0.5 h. After that, 28.775 mg of NHS were added and properly stirred at 35 °C for 3 h. A quantity of 425 mg of NH_2_-PEG_2K_-NH_2_ that dissolved with 25 mL of DMSO and 3 drops of TEA was added and appropriately stirred at 35 °C for 3 days. After that, equal proportions of purified water were added, and the mixed solution was dialyzed against water for 1 day in dialysis tubing (MWCO 2000). Then, the intermediate product was freeze-dried and dissolved with 15 mL of DMSO. Quantities of 38.65 mg of lipoic acid, 59.875 mg of EDCI, and 38.125 mg of DMAP were added and were stirred properly at 35 °C for 3 days. After the reaction, equal proportions of purified water were added, and the mixed solution was dialyzed against water for 3 days in dialysis tubing (MWCO 2000). Then, a yellow flocculation product was obtained after freeze-drying.

(4)Preparation of AuNSs@PEG and AuNSs@FA

AuNSs@PEG: La-PEG (100 µL, 10 mg/mL) was added to 5 mL of AuNSs solution and was stirred overnight at a temperature of 25 °C.

AuNSs@FA: La-PEG (75 µL, 10 mg/mL) and La-PEG-FA (25 µL, 10 mg/mL) were added to 5 mL of AuNSs solution in turn, and the mixture was stirred overnight at a temperature of 25 °C.

### 3.3. Nanoparticle Characterization

Volumes of 1000 μL of AuNSs, AuNSs@PEG, and AuNSs@FA were added to a centrifuge tube ([Au] concentration: 0.4 mM). A Zetasizer Nano ZS90 (Malvern Instrument, Malvern, UK) was used to measure the nanoparticles’ size and zeta potential at a temperature of 25 °C, and their ultraviolet absorption spectra were obtained from a UV-Vis spectrophotometer (UV-1700; Shimadzu, Kyoto, Japan).

Stability test: 200 μL of AuNSs@PEG and AuNSs@FA were added into 1800 μL of PBS or RPMI ([Au] concentration: 0.4 mM). Then, the solution was divided into two parts: one was preserved in a refrigerator at 4 °C, and the other one was preserved on a shaking table at 37 °C. The Zetasizer Nano ZS90 instrument was used to measure their size after one week.

### 3.4. Photothermal Conversion Efficiency

A volume of 1 mL of AuNSs@PEG or AuNSs@FA at Au concentrations of 25, 50, 75, 100, and 150 mg/L were contained in a 2.0-milliliter centrifuge tube and irradiated by an 808-nanometer laser (1.5 W/cm^2^) for 10 min. The temperature changes were monitored by a thermal image camera in the process. Moreover, the temperature change of AuNSs@PEG or AuNSs@FA alone or combined with CAP at an Au concentration of 100 mg/L at different power densities (0.5, 1, 1.5, 2 W/cm^2^) with the 808-nanometer laser was also measured. After that, the cyclic curves and photostability were measured. The AuNSs@PEG or AuNSs@FA at an Au concentration of 100 mg/L were added into a 2.0-milliliter centrifuge tube and irradiated by the 808-nanometer laser (1.5 W/cm^2^) for 5 min and cooled for 20 min. The temperature changes were monitored by the thermal image camera in the process. In combination with the CAP groups, AuNSs@PEG or AuNSs@FA were treated with CAP (industrial-grade Argon (Ar) was used as a carrier gas; the flow rate was 2 L/min and treatment time was 1.5 min) before being irradiated by NIR.

### 3.5. In Vitro Experiments

(1)Hemolysis test

The fresh blood samples from BALB/c mice were centrifuged for 10 min at 5000 rpm, then washed 3 times with PBS until the supernatant was clarified to obtain red blood cells (RBCs). A volume of 3 mL of centrifuged erythrocytes was mixed with 11 mL of saline to prepare the stock dispersion. A volume of 100 μL of stock dispersion was added to 1 mL of buffers containing AuNSs@PEG or AuNSs@FA at Au concentrations of 0.2, 0.4, 0.6, 0.8, and 1.0 mM. After incubation at 37 °C for 3 h, the mixtures were centrifuged at 13,000 rpm for 15 min. The absorbance of the supernatant at 540 nm was measured by UV-vis. The following formula was used to calculate the hemolysis rate:

Hemolysis percentages (%) = (A_S_ − A_N_)/(A_P
_ − A_N_) × 100%
(1)
A_S_, A_N_, and A_P_ are the absorbances of AuNSs@PEG or AuNSs@FA, saline, and deionized water, respectively. Saline and deionized water were used as negative control and positive control, respectively.

(2)Cell preparation

L929 cells were maintained in a DMEM culture medium containing 10% FBS, 1% pyruvic acid sodium salt, and 1% penicillin-streptomycin. 4T1 and 231 cells were grown in RPMI added with 10% FBS, 1% pyruvic acid sodium salt, and 1% penicillin-streptomycin. All cells were cultured at 37 °C in a humidified incubator with 5% CO_2_. The cytotoxicity of AuNSs@PEG or AuNSs@FA was evaluated by these cells.

(3)Cytotoxicity test In vitro

For evaluating the biocompatibility of AuNSs@PEG or AuNSs@FA, L929 cells were used for an MTT assay. L929 cells were maintained in 100 µL of DMEM in 96-well plates (8 k per well) for 24 h, then incubated with AuNSs@PEG or AuNSs@FA at Au concentrations of 0.2, 0.4, 0.6, 0.8, 1.0 mM, at 37 °C with 5% CO_2_ for 24 h. After that, 10 μL of MTT was added. After culturing for 4 h, cells were washed with PBS, then dissolved in 100 µL of DMSO. Using an automatic enzyme immunoassay instrument (Multiskan Go, Thermo Scientific, Waltham, MA, USA), the absorbance of each well in the plates was measured at 490 nm.

Moreover, the therapeutic effect in vitro of the AuNSs@PEG or AuNSs@FA on cancer cells was also assessed. The 231 cells were grown in 100 µL of RPMI in 96-well plates (7 k per well) for 12 h. Then, the cells were treated with AuNSs@PEG, AuNSs@FA, AuNSs@PEG+CAP, AuNSs@PEG+1.5 W/cm^2^, AuNSs@PEG+0.5 W/cm^2^, AuNSs@PEG+CAP+1.5 W/cm^2^, AuNSs@PEG+CAP+0.5 W/cm^2^, AuNSs@FA+CAP, AuNSs@FA+1.5 W/cm^2^, AuNSs@FA+0.5 W/cm^2^, AuNSs@FA+CAP+1.5 W/cm^2^, and AuNSs@FA+CAP+0.5 W/cm^2^ at Au concentrations of 0.2, 0.4, 0.6, 0.8, and 1.0 mM. Cell viability was evaluated using the above-mentioned MTT. Industrial-grade Ar was used as a carrier gas in CAP. The flow rate was 2 L/min, and the treatment time was 1.5 min.

A live/dead assay was also used for assessing the therapeutic efficacy in vitro of AuNSs@PEG or AuNSs@FA. The 231 cells were maintained in 0.8 mL of RPMI in 24-well plates (12 k per well) overnight. Then, they were treated with AuNSs@PEG, AuNSs@FA, AuNSs@PEG+CAP, AuNSs@PEG+1.5 W/cm^2^, AuNSs@PEG+0.5 W/cm^2^, AuNSs@PEG+CAP+1.5 W/cm^2^, AuNSs@PEG+CAP+0.5 W/cm^2^, AuNSs@FA+CAP, AuNSs@FA+1.5 W/cm^2^, AuNSs@FA+0.5 W/cm^2^, AuNSs@FA+CAP+1.5 W/cm^2^, and AuNSs@FA+CAP+0.5 W/cm^2^ for 24 h at an Au concentration of 0.4 mM. After the treatments, cells were washed with PBS twice, and a fluorescence microscope was used to evaluate the cells that were incubated with 100 µL of AM/PI cocktail for 20 min. Industrial-grade Ar was used as a carrier gas in CAP. The flow rate was 2 L/min, and the treatment time was 1.5 min.

## 4. Conclusions

In summary, we used CAP combined with the nano-formulation AuNSs@FA to achieve precise treatment of the tumor site. Supplementing lower power irradiation treatment with CAP yielded similar therapeutic effects as higher power irradiation treatment alone. Furthermore, lower power irradiation treatment results in less or no damage to the skin around the tumor. The selective killing effect was proved after its feasibility assessment in vitro and in vivo studies. Relative to increasing PTA accumulation or making self-regulating PTAs, using CAP as a supplementary procedure to enhance the selective PTT for cancer is simpler and easier to transform. We thus believe that CAP can potentially be extended into other therapeutic applications.

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
