# Peer review of "Cold Atmospheric Plasma Activates Selective Photothermal Therapy of Cancer"

_molecules, 2022, doi:10.3390/molecules27185941_

Round 1

Reviewer 1 Report

In this manuscript, Qin et al. used cold atmospheric plasma (CAP) as a supplementary procedure to enhance the selective PTT of cancer of the classical photothermic agent's gold nanostars (AuNSs). Both in vitro and in vivo experiments have proved that the combination therapy can achieve rapid tumor suppression in the early stage of treatment and reduces the side effects on surrounding normal tissues compared to PTT alone. Overall, most parts of this manuscript are well-organized. However, some revisions should be made before publication:
1. The targeting of folic acid needs to be verified by cell imaging experiments.
2. The effect of combination therapy on normal skin needs to be verified.
3. The scale bar in figure 1b is not clear. Please correct them.

4. The fonts in some figures are too small to read. Such as Fig. 5.
5. Some related articles should be referred (Angew. Chem. Int. Ed., 2022, 61, e202115939; Inorg. Chem., 2022, 61, 9328-9338; Chem. Soc. Rev., 2022, 51, 6126-6176; Dalton Trans. 2020, 49, 17772-17778.).

Author Response

Dear Editor,

Thank you and the reviewers very much for taking the time and effort to review the manuscript. Our manuscript “Cold Atmospheric Plasma Activate Selective Photothermal Therapy of Cancer (Manuscript ID: molecules-1861085)” has been carefully revised according to your instructions and comments, and point-by-point responses to your and the reviewer's comments are provided below. We have used the “Track Changes” function to revise the revised manuscript.

Reviewer 1

  1. The targeting of folic acid needs to be verified by cell imaging experiments.

Answer: Thank you for your suggestion. The targeting of folic acid has been proved in much research[1-4]. so here we do not repeat the test. Moreover, from the photothermal curve, AuNSs@FA and AuNSs modified without FA have similar photothermal conversion capability, but compared with AuNSs modified without FA, AuNSs@FA has greater cell killing in the in vitro test of this manuscript, illustrating AuNSs@FA is taken up more by cells and have targeting properties.

References:

1 Z. Zhou, Y. Wang, F. Peng, F. Meng, J. Zha, L. Ma, Y. Du, N. Peng, L. Ma, Q. Zhang, L. Gu, W. Yin, Z. Gu, C. Tan, Intercalation-Activated Layered MoO(3) Nanobelts as Biodegradable Nanozymes for Tumor-Specific Photo-Enhanced Catalytic Therapy, Angew Chem Int Ed Engl 61(16) (2022) e202115939.

2 X. Zhao, X. He, A. Hou, C. Cheng, X. Wang, Y. Yue, Z. Wu, H. Wu, B. Liu, H. Li, J. Shen, C. Tan, Z. Zhou, L. Ma, Growth of Cu(2)O Nanoparticles on Two-Dimensional Zr-Ferrocene-Metal-Organic Framework Nanosheets for Photothermally Enhanced Chemodynamic Antibacterial Therapy, Inorg Chem 61(24) (2022) 9328-9338.

3 J.H. Qin, H. Zhang, P. Sun, Y.D. Huang, Q. Shen, X.G. Yang, L.F. Ma, Ionic liquid induced highly dense assembly of porphyrin in MOF nanosheets for photodynamic therapy, Dalton Trans 49(48) (2020) 17772-17778.

4 T. Hu, Z. Gu, G.R. Williams, M. Strimaite, J. Zha, Z. Zhou, X. Zhang, C. Tan, R. Liang, Layered double hydroxide-based nanomaterials for biomedical applications, Chem Soc Rev 51(14) (2022) 6126-6176.

  1. The effect of combination therapy on normal skin needs to be verified.

Answer: Thank you for your suggestion. We have verified the effect of combination therapy on normal skin.

Revision: As shown in Figure 8c, the H&E slice section of the skin with AuNSs@FA+1.5 W/cm2 treatment have inflammation, but AuNSs@FA+CAP+1.0 W/cm2 treatment group have no inflammation in the H&E slice section of the skin, illustrating combination of PTT and CAP can reduce power and inflammation to skin tissue to achieve selective treatment.

Figure 8: (c) H&E slice section of the skin with AuNSs@FA+1.5 W/cm2 and AuNSs@FA+CAP+1.0 W/cm2 treatments.

  1. The scale bar in figure 1b is not clear. Please correct them.

Answer: Sorry. This is our negligence, and we have added the scale bar to Figure 1b.

  1. The fonts in some figures are too small to read. Such as Fig. 5.

Answer: I am sorry for that. We have revised the font size for easy reading.

  1. Some related articles should be referred (Angew. Chem. Int. Ed., 2022, 61, e202115939; Inorg. Chem., 2022, 61, 9328-9338; Chem. Soc. Rev., 2022, 51, 6126-6176; Dalton Trans. 2020, 49, 17772-17778.)

Answer: Thank you very much for your suggestion. These references have been cited in this manuscript.

Reviewer 2 Report

The manuscript submitted by Dr. Qin et al. deals with an enhanced effect of cold atmospheric plasma on photothermal cancer therapy using Au nanoparticles. This is an important and interesting topic in nanomaterials and photothermal therapy, so that it is fully expected that the work can contribute to develop such research fields. Therefore, I can recommend the manuscript to be published in the journal, Molecules, after appropriate corrections. 

I feel there are many misprints and unclear explanations in the manuscript.

For example,

Line 102:  and The mixed   and the mixed

A scale bar almost disappears in Figure 1b.

Line 242  the word “UV” may be wrong in the caption of Figure 1c.     

Line 242  SD deviation    SD

Line 252-   More reasonable reason is necessary for the enhancement of photothermal effect with CAP, because this is a key in this full paper.

Line 251:  Fig.2e   Fig.2c

Line 257:  temperature       temperatures        

Line 258:  Fig.3e and 3f    Figs 3c and 3f

Line 258:  What is “the corresponding irradiation parameters”?  Dose it mean “temperature”?

Line 260:  stability    stabilities

Line 268:  different Au concentrations of Au     different concentrations of Au

Legends in the figures should be removed in in Figures 4abs and 5de.

Line 277:  AuNSs@FA+CAP     AuNSs@PEG+CAP

A name with unit of the horizonal axis is missing in Figs. 2c, 2f, 3c, 3f, 7d

The descriptions of the caption in Figures 5 and 6, 7 are too simple to understand the figures. More polite and detailed explanation is necessary in the captions.

Lines 314 and 326:  Underlines should be removed.  

Line 337-    The result of AuNSs without FA should be shown in Figure 7a, as the authors mention “Compared with the group of AuNSs modified without FA” in the text.

It is unknown which part in the photo corresponds to which organ and tumor specifically in Figure 7b..  

Line 364:  that Tumor volume       that tumor volume

The results of AuNSs@PEG+CAP should be shown in Figures 7c, 7d, 7e and 7f, because it is interesting and important to see the additional temperature increase by the CAP treatment and compare the CAP effect not only in AuNSs@FA but also in AuNSs@PEG systems in vivo.

 A description about the coincidental overlapping of the plots of AuNSs@FA+1.5 W/cm3 with those of AuNSs@FA+CAP +1.0 W/cmin Figure 8a should be added in the text. 

What are +L1 and +L2 in the legend of Figure 8a?

Lines 386, 387:  What dose “PTAs” mean?

Character size in the legend in figures are too small to see. Instead of putting the legend into the figures, proper descriptions should be added in the caption of Figures 1a, 1c, 2abcdef, 3abcdef, 5abcf, 6abcd.    

Author Response

Dear Editor,

Thank you and the reviewers very much for taking the time and effort to review the manuscript. Our manuscript “Cold Atmospheric Plasma Activate Selective Photothermal Therapy of Cancer (Manuscript ID: molecules-1861085)” has been carefully revised according to your instructions and comments, and point-by-point responses to your and the reviewer's comments are provided below. We have used the “Track Changes” function to revise the revised manuscript.

Reviewer 2

I feel there are many misprints and unclear explanations in the manuscript.

For example,

  1. Line 102: and The mixed→and the mixed

Answer: Sorry, This is our negligence, and we have changed “and The mixed” to “and the mixed”.

  1. A scale bar almost disappears in Figure 1b.

Answer: Sorry. This is our negligence, and we have added the scale bar to Figure 1b.

  1. Line 242 the word “UV” may be wrong in the caption of Figure 1c.

Answer: Thank you. It is wrong and we have changed “UV” to “UV-Vis”.

  1. Line 242 SD deviation→SD

Answer: Sorry, This is our negligence, and we have changed “SD deviation” to “SD”.

  1. Line 252-More reasonable reason is necessary for the enhancement of photothermal effect with CAP, because this is a key in this full paper.

Answer: Thank you very much for your suggestion. CAP could emit high energy and charged particles on the treated surface, resulting in the breakage of the covalent bonds and initiating many chemical reactions[5, 6]. Through further literature review, we speculated that CAP activates selective PTT of cancer not by increasing the photothermal conversion efficiency of the photothermal agents. Selectivity is enhanced by increasing oxidative stress levels to cause cell toxicity and death, and T cell-mediated immune response in tumor cells to reduce tumor activity[7, 8]. The combination of PTT with CAP at lower power achieved a similar killing effect to that of high power.

References:

5 M. Wang, P. Favi, X. Cheng, N.H. Golshan, K.S. Ziemer, M. Keidar, T.J. Webster, Cold atmospheric plasma (CAP) surface nanomodified 3D printed polylactic acid (PLA) scaffolds for bone regeneration, Acta Biomater 46 (2016) 256-265.

6 J. Tornin, C. Labay, F. Tampieri, M.P. Ginebra, C. Canal, Evaluation of the effects of cold atmospheric plasma and plasma-treated liquids in cancer cell cultures, Nat Protoc 16(6) (2021) 2826-2850.

7 X. Dai, K. Bazaka, E.W. Thompson, K.K. Ostrikov, Cold Atmospheric Plasma: A Promising Controller of Cancer Cell States, Cancers (Basel) 12(11) (2020).

8 C. Bengtson, A. Bogaerts, On the Anti-Cancer Effect of Cold Atmospheric Plasma and the Possible Role of Catalase-Dependent Apoptotic Pathways, Cells 9(10) (2020).

  1. Line 251: Fig.2e→Fig.2c

Answer: I am sorry for that. We have changed “Fig. 2e” to “Fig. 2c”.

  1. Line 257: temperature→temperatures

Answer: Sorry, This is our negligence, and we have changed “temperature” to “temperatures”.

  1. Line 258: Fig.3e and 3f→Figs 3c and 3f

Answer: I am sorry for that. We have changed “Fig. 3e and 3f” to “Fig. 3c and 3f”.

  1. Line 258: What is “the corresponding irradiation parameters”? Dose it mean “temperature”?

Answer: Thank you. It should be “temperature”, and we have revised this part.

  1. Line 260: stability→stabilities

Answer: Sorry, This is our negligence, and we have changed “stability” to “stabilities”.

  1. Line 268: different Au concentrations of Au→different concentrations of Au

Answer: I am sorry for that. We have changed “different Au concentrations of Au” to “different concentrations of Au”.

  1. Legends in the figures should be removed in Figures 4abs and 5de.

Answer: Thank you for your suggestion. We have deleted them and put them in Supporting Information.

  1. Line 277: AuNSs@FA+CAP→AuNSs@PEG+CAP

Answer: Sorry, This is our negligence, and we have changed “AuNSs@FA+CAP” to “AuNSs@PEG+CAP”.

  1. A name with unit of the horizonal axis is missing in Figs. 2c, 2f, 3c, 3f, 7d

Answer: Thank you for your question. We have added the horizontal axis in Figs. 2c, 2f, 3c, 3f, 7d.

  1. The descriptions of the caption in Figures 5 and 6, 7 are too simple to understand the figures. More polite and detailed explanation is necessary in the captions.

Answer: Thank you for your suggestion. We have revised this part.

Revision: Figure 5: Stability of AuNSs@PEG and AuNSs@FA cultured in PBS and RPMI at 4°C (a) or 37°C (b) with 12h. (c) Stability of AuNSs@PEG and AuNSs@FA cultured in PBS within 1 week at 4°C and 37°C. (d) Cell viability of L929 cells in the different Au concentrations of AuNSs@PEG and AuNSs@FA. (e) H&E slice selection section of the major organs (heart, liver, spleen, lung, and kidney) on day 10.

Figure 6: Cell viability of 231 cells after NIR irradiation at 0.5 W/cm2 (a) and 1.5 W/cm2 (b) combined with CAP treatment. Cell viability of 231 cells of AuNSs@PEG (c) and AuNSs@FA (d) after treatment with different laser irradiation (0.5 W/cm2 or 1.5 W/cm2) and CAP. (e) LIVE/DEAD assay of 231 cells with the different treatments in the Au concentration of 0.4 mM.

Figure 7: (a, b) Biodistribution of AuNSs@FA and AuNSs@PEG in tumor-bearing nude mice at different points of time (2, 6, 12, 24, 48 hours), and tumor and the major organs collected at 6 h and 48 h point. Infrared thermal images in vivo (c, e) and the temperature evolution curves (d, f) treated with PBS, AuNSs@FA, AuNSs@FA+CAP, and AuNSs@PEG under 808 nm laser-irradiation at 1.0 and 1.5 W/cm2 respectively within 5 minutes.

  1. Lines 314 and 326: Underlines should be removed.

Answer: Thank you for your suggestion. We have removed the underlines.

  1. Line 337-The result of AuNSs without FA should be shown in Figure 7a, as the authors mention “Compared with the group of AuNSs modified without FA” in the text.

Answer: Thank you very much for your suggestion. AuNSs@FA was modified with La-PEG-FA, and AuNSs@PEG was modified with La-PEG and represented no modification of FA in this manuscript.

  1. It is unknown which part in the photo corresponds to which organ and tumor specifically in Figure 7b.

Answer: Thank you. We have added organ and tumor labels in Figure 7b.

  1. Line 364: that Tumor volume→that tumor volume

Answer: Sorry, This is our negligence, and we have changed “that Tumor volume” to “that tumor volume”.

  1. The results of AuNSs@PEG+CAP should be shown in Figures 7c, 7d, 7e and 7f, because it is interesting and important to see the additional temperature increase by the CAP treatment and compare the CAP effect not only in AuNSs@FA but also in AuNSs@PEG systems in vivo.

Answer: Thank you for your suggestion. CAP activates selective PTT of cancer not by increasing the additional temperature increase. Selectivity is enhanced by increasing oxidative stress levels to cause cell toxicity and death, and T cell-mediated immune response in tumor cells to reduce tumor activity. As shown in Figures 7c, 7d, 7e, and 7f, AuNSs@FA+CAP compared with AuNSs@FA have no obvious temperature change.

  1. A description about the coincidental overlapping of the plots of AuNSs@FA+1.5 W/cm3 with those of AuNSs@FA+CAP +1.0 W/cm3 in Figure 8a should be added in the text.

Answer: Thank you very much for your suggestion. This is our negligence. There was a paste error in the process of processing, and we have corrected it in Figure 8a.

  1. What are +L1 and +L2 in the legend of Figure 8a?

Answer: Thank you for your question. L1 and L2 mean 1.5 W/cm2 and 1.0 W/cm2 respectively, as we have remarked in Figure 8a.

  1. Lines 386, 387: What dose “PTAs” mean?

Answer: Thank you for your question. “PTAs” means “photothermal agents”.

  1. Character size in the legend in figures are too small to see. Instead of putting the legend into the figures, proper descriptions should be added in the caption of Figures 1a, 1c, 2abcdef, 3abcdef, 5abcf, 6abcd.

Answer: Thank you for your suggestion. We have revised the font size to easy reading and added proper descriptions. 

Author Response

Dear Editor,

Thank you and the reviewers very much for taking the time and effort to review the manuscript. Our manuscript “Cold Atmospheric Plasma Activate Selective Photothermal Therapy of Cancer (Manuscript ID: molecules-1861085)” has been carefully revised according to your instructions and comments, and point-by-point responses to your and the reviewer's comments are provided below. We have used the “Track Changes” function to revise the revised manuscript.

Reviewer 3

  1. Line 14. Introduce a better background of this study.

Answer: Thanks. Based on your question, we have revised this part.

Revision: Due to the body's systemic distribution of photothermal agents (PTAs) and non-precision exposure to lasers, photothermal therapy (PTT) is challenging to treat the tumor site precisely. Exploring PTT with high selectivity and precise treatment is still important to raise the survival rate of cancer patients and lower the likelihood of adverse effects on other body sections.

  1. Line 17-19. Improve the sentence, it is difficult to read.

Answer: Thank you for your suggestion. We have revised the sentence for easy reading.

Revision: In vitro experiments, CAP decreases the effective power of PTT: the combination of PTT with CAP at lower power has similar cytotoxicity to that at higher power.

  1. Line 46-47, 49-51. Why is CAP more selective for cancer cells relative to healthy cells? Explain in greater depth the generation of CAP and its effect and place more bibliographical references on studies that use CAP and CAP-PTT.

Answer: Thank you very much for your suggestion. We have revised this part.

Revision: The anticancer selectivity is related to the concentration of reactive oxygen and nitrogen species (RONS) generated by CAP[28]. CAP can target and kill cancer cells by increasing oxidative stress levels. RONS also can induce cancer immunogenic cell death (ICD) and then release tumor-associated antigens (TAAs). In tumor-draining lymph nodes, mature dendritic cells (DCs) give antigen peptides to T cells, producing cytotoxic T cells. The subsequent T-cell-mediated immune response can inhibit tumor recurrence and growth[29]. In addition, higher aquaporin density on the cell membrane causes a higher inflow of hydrogen peroxide to kill cancer cells[8, 30, 31].

References:

8 T. Hu, Z. Gu, G.R. Williams, M. Strimaite, J. Zha, Z. Zhou, X. Zhang, C. Tan, R. Liang, Layered double hydroxide-based nanomaterials for biomedical applications, Chem Soc Rev 51(14) (2022) 6126-6176.

28 J. Tornin, C. Labay, F. Tampieri, M.P. Ginebra, C. Canal, Evaluation of the effects of cold atmospheric plasma and plasma-treated liquids in cancer cell cultures, Nat Protoc 16(6) (2021) 2826-2850.

29 G. Chen, Z. Chen, Z. Wang, R. Obenchain, D. Wen, H. Li, R.E. Wirz, Z. Gu, Portable air-fed cold atmospheric plasma device for postsurgical cancer treatment, Sci Adv 7(36) (2021) eabg5686.

30 X. Dai, K. Bazaka, E.W. Thompson, K.K. Ostrikov, Cold Atmospheric Plasma: A Promising Controller of Cancer Cell States, Cancers (Basel) 12(11) (2020).

31 C. Bengtson, A. Bogaerts, On the Anti-Cancer Effect of Cold Atmospheric Plasma and the Possible Role of Catalase-Dependent Apoptotic Pathways, Cells 9(10) (2020).

  1. Line 56-61. This paragraph should be removed because the results are not in the "Introduction" section.

Answer: Thank you very much for your suggestion. We have deleted the results part.

  1. Scheme 1. I suggest incorporating an illustration of CAP on the tumor surface.

Answer: Thank you for your suggestion. We have incorporated an illustration of CAP on the tumor surface.

Revision:

  1. Improve the resolution of all images.

Answer: Thank you for your suggestion. We have improved the resolution of all images.

  1. Line 184 incorporates the name of the tumor cell line used.

Answer: Thank you for your suggestion. We have added the name of the tumor cell line used. 4T1 (mouse breast cancer cells) was used to establish the tumor model.

  1. Lines 237-238, 254, 340. Place bibliographical references.

Answer: Thank you for your suggestion. We have added the references.

Revision: Line 237-238: Moreover, the zeta potential change of AuNSs further proves the success of FA modification[38] (Fig. 1d).

Line 254: CAP is an effective material surface modification tool. It could emit high energy and charged particles on the treated surface, resulting in the breakage of the covalent bonds and initiating many chemical reactions[41].

Line 340: Compared with the group of AuNSs modified without FA, the tumor site in the group of AuNSs@FA exhibited stronger fluorescence at 6 h, illustrating the modification of FA can improve the tumor-targeting ability of nanoparticles[47].

References:

38 L. Zhao, X. Wang, H. Lou, M. Jiang, X. Wu, J. Qin, J. Zhang, X. Guan, W. Li, W. Zhang, J. Ma, Buffet-style Cu(II) for enhance disulfiram-based cancer therapy, J Colloi

41 M. Adhikari, N. Kaushik, B. Ghimire, B. Adhikari, S. Baboota, A.A. Al-Khedhairy, R. Wahab, S.J. Lee, N.K. Kaushik, E.H. Choi, Cold atmospheric plasma and silymarin nanoemulsion synergistically inhibits human melanoma tumorigenesis via targeting HGF/c-MET downstream pathway, Cell Commun Signal 17(1) (2019) 52.Sci 624 (2022) 734-746.

47 F. Yugui, H. Wang, D. Sun, X. Zhang, Nasopharyngeal cancer combination chemoradiation therapy based on folic acid modified, gefitinib and yttrium 90 co-loaded, core-shell structured lipid-polymer hybrid nanoparticles, Biomed Pharmacother 114 (2019) 108820.

  1. Figures 2c and 2d place Y axis title.

Answer: Thank you for your suggestion. We have added this part.

  1. Line 247, 251, 256, 258. Check that the numbering described in the manuscript is consistent with the numbering in figures 2 and 3, respectively.

Answer: Thank you for your questions. We have checked and changed the numbering described in the manuscript in Figures 2 and 3.

  1. Figure 5. Increasing the size of the chart boxes to improve resolution

Answer: Thank you for your suggestion. We have changed the size of the chart boxes and improved the resolution.

  1. You consider that the presence of ROS with and without CAP should have been quantified to determine if the technique used is only PTT or PTT/Photodynamic Therapy. It would be appropriate to place a discussion paragraph on this point.

Answer: Thank you very much for your questions. The existence of ROS in CAP has been demonstrated in a large number of works of literature, so here we do not repeat the test. The technique used is only PTT combined CAP. CAP can increase oxidative stress levels to cause cell toxicity and death. AuNSs@FA only has the function of targeting tumors and providing heat. Photodynamic Therapy eliminates local cancers through the highly cytotoxic ROS. Photosensitizers generate ROS by producing radicals and radical ions and generating highly reactive singlet oxygen (1O2).

  1. Figure 8 a. Specify the letters L1 and L2 in the title of the figure.

Answer: Thank you for your suggestion. L1 and L2 mean 1.5 W/cm2 and 1.0 W/cm2 respectively, as we have remarked in Figure 8a.

  1. I suggest using an ANOVA and Tukei analysis to find if the differences between pairs are significant.

Answer: Thank you for your suggestion. We have analyzed the significant differences in Figures.

Reviewer 4 Report

Dear Editor,

Journal: Molecules

Manuscript ID: molecules-1861085

The manuscript titled Cold Atmospheric Plasma Activate Selective Photothermal Therapy of
Cancer is novel, interesting work and can be published with answering few queries and modification given below

1)    The author describes the CAP enhances the photothermal please provide the mechanism of CAP with photothermal

2)    Why gold nanostars were chosen as photothermal agents rather than rods which as higher cellular uptake

3)    The Conjugation of gold nanostars with folic acid is not  characterised for the FTIR spectra

4)    Figure 1b TEM image scale bar is not visible

5)    Please clarify the sentence 253 bonds are broken what it refers ?

6)    Please replace graph 8a with quality image

7)    Mechanism of RNS and ROS with CAP just mention in one sentence ?

Regards

Lakshmi narashimhan R

Author Response

Dear Editor,

Thank you and the reviewers very much for taking the time and effort to review the manuscript. Our manuscript “Cold Atmospheric Plasma Activate Selective Photothermal Therapy of Cancer (Manuscript ID: molecules-1861085)” has been carefully revised according to your instructions and comments, and point-by-point responses to your and the reviewer's comments are provided below. We have used the “Track Changes” function to revise the revised manuscript.

Reviewer 4

1) The author describes the CAP enhances the photothermal please provide the mechanism of CAP with photothermal

Answer: Thank you for your suggestion. We have revised this part.

Revision: Cold atmospheric plasma (CAP) is a highly energized gas composed of numerous biologically active factors where the ion temperature is close to room temperature. They may act synergistically on the treated materials and tissues and may have various biologic effects[6, 19-21]. CAP has numerous biologically active components, of which the basic components are various reactive oxygen species (ROS) and reactive nitrogen species (RNS) formed in reaction with molecules (oxygen, nitrogen, and water) present in the ambient air[22]. Research suggested that CAP provides a selectively targeting cancer cells effect, as cancer cells are more sensitive to CAP treatment than healthy cells[22-27]. The anticancer selectivity is related to the concentration of reactive oxygen and nitrogen species (RONS) generated by CAP[28]. CAP can target and kill cancer cells by increasing oxidative stress levels. RONS also can induce cancer immunogenic cell death (ICD) and then release tumor-associated antigens (TAAs). In tumor-draining lymph nodes, mature dendritic cells (DCs) give antigen peptides to T cells, producing cytotoxic T cells. The subsequent T-cell-mediated immune response can inhibit tumor recurrence and growth[29]. In addition, higher aquaporin density on the cell membrane causes a higher inflow of hydrogen peroxide to kill cancer cells[8, 30, 31]. However, the direct killing ability of the tumor of CAP is weak, and a supplementary procedure or a precautionary measure may be its primary application strategy[20, 32]. Therefore, we consider whether realizing the selective for PTT of cancer used CAP as a supplementary procedure. The combination of PTT with CAP at lower power achieved a similar killing effect to that of high power.

2) Why gold nanostars were chosen as photothermal agents rather than rods which as higher cellular uptake

Answer: Good question. Theoretically, gold nanostars and gold rods both can be used for CAP to activate selective PTT of cancer. They both have good photothermal conversion ability as PTAs. AuNSs is more green and convenient to prepare, and we choose them as a model drug material.

3) The Conjugation of gold nanostars with folic acid is not characterised for the FTIR spectra

Answer: Thank you for your suggestion. FA is bound to the surface of AuNSs through the S-Au bond, which has been proved in a large number of literature, so here we do not repeat the test.

4) Figure 1b TEM image scale bar is not visible

Answer: Sorry. This is our negligence, and we have added the scale bar to figure 1b.

5) Please clarify the sentence 253 bonds are broken what it refers ?

Answer: Thank you for your suggestion. CAP is an effective material surface modification tool. The sentence 253 “bonds are broken” refers to the breakage of the covalent bonds and initiation of many chemical reactions to infer the reasons for the increase of material temperature after CAP treatment.

6) Please replace graph 8a with quality image

Answer: Thank you. We have improved the resolution of Figure 8a.

7) Mechanism of RNS and ROS with CAP just mention in one sentence ?

Answer: Thank you for your question. We have revised this part.

Revision: CAP has numerous biologically active components, of which the basic components are various reactive oxygen species (ROS) and reactive nitrogen species (RNS) formed in reaction with molecules (oxygen, nitrogen, and water) present in the ambient air[22]. Research suggested that CAP provides a selectively targeting cancer cells effect, as cancer cells are more sensitive to CAP treatment than healthy cells[22-27]. The anticancer selectivity is related to the concentration of reactive oxygen and nitrogen species (RONS) generated by CAP[28]. CAP can target and kill cancer cells by increasing oxidative stress levels. RONS also can induce cancer immunogenic cell death (ICD) and then release tumor-associated antigens (TAAs). In tumor-draining lymph nodes, mature dendritic cells (DCs) give antigen peptides to T cells, producing cytotoxic T cells. The subsequent T-cell-mediated immune response can inhibit tumor recurrence and growth[29]. In addition, higher aquaporin density on the cell membrane causes a higher inflow of hydrogen peroxide to kill cancer cells[8, 30, 31].